# OpenReview forum: "Steered LLM Activations are Non-Surjective"
_ICLR.cc/2026/Workshop/Sci4DL — Sci4DL 2026_

### Official Review · Reviewer_Zop1 · 2026-02-25

**Fit:** 2
**Significance:** 1
**Confidence:** 2

**Summary:**

This paper builds upon Nikolaou et al's recent result that LLMs are injective to further claim that steered LLM activations are non-surjective i.e., when applying a steering vector to an input's activations to cause some change in output behaviour, these new activations cannot be obtained with any prompt. They test this theory for both random steering vectors and "meaningful" steering vectors which are meant to represent a particular property. In both cases, the authors claim that collisions "almost surely" never happen. The content of the paper relies heavily on the previous results from Nikolaou et al.

**Strengths:**

- The idea is interesting, timely, and worth exploring. The distinction between white-box steerability and black-box prompting is worth stating and investigating further.

**Suggestions:**

- This reads like it should probably be a full paper but on principle, I am evaluating it in its 4-page state. The majority of the paper is spent recounting the results from Nikolaou et al, meanwhile their own experimental setup is left to the appendix. One cannot interpret the results without also reading the appendix, which, in my view, amounts to artificially extending the page count.
- The biggest weakness in this paper is that the claims are too big given the results. To say that "almost surely, no prompt can reproduce the same internal behaviour reproduced by steering" requires major theoretical results which simply are not there. The proofs for theorem 2.1 are left to the appendix and are not particularly formal or convincing. They defer substantially to the results from Nikolaou et al (again) e.g., "We replace X with Y in Nikolaou et al's proof". The reader should not be expected to read another full paper plus the entire appendix to understand the results of this 4-page paper.
- Figure 1: The authors write that "LLMs are injective... this property implies existence of holes: regions that do not map back to any prompt", which is not the case. If something is bijective, then there are no holes. Injective while not surjective would imply there are holes, but that is not what's written in the figure caption. This is probably just a typo but please fix it.

---

### Official Review · Reviewer_pMcD · 2026-02-26

**Fit:** 2
**Significance:** 2
**Confidence:** 3

**Summary:**

SUMMARY
--------
This paper asks a question that is deceptively simple but surprisingly underexplored:
when you apply activation steering to an LLM, do the resulting internal states actually
correspond to anything a real text prompt could produce? The authors cast this as a
surjectivity problem and prove, under practical assumptions, that the answer is almost
surely no. The argument builds on the real-analyticity of transformers and an injectivity
result from Nikolaou et al. (2025) to show that steering vectors move the residual stream
off the manifold of prompt-reachable activations. Two empirical paradigms are used to
validate this: SIPIT inversion (which successfully recovers natural activations but fails
on steered ones) and many-shot ICL (which shows that even as jailbreak success rates rise,
activations diverge further from steered ones rather than converging). The paper concludes
by arguing that white-box steerability and black-box prompt exploitability should be
treated as distinct quantities in safety evaluations, a point illustrated with a simple
"Here attack" that matches steering in attack success rate while operating through a
completely different internal mechanism.

**Strengths:**

STRENGTHS
----------
1. The surjectivity framing is genuinely novel and useful. The question of whether
steered activations are prompt-reachable has been lurking in the background of the
interpretability and safety literature for a while, but no one had turned it into a
precise formal question before. Doing so is a real contribution — it gives the community
a well-defined concept to reason about and opens the door to follow-up work that could
not have been framed without this vocabulary.

2. The experimental design is careful and honest. The key move of using SIPIT inversion
on natural activations as a baseline before testing steered ones is exactly right. It
separates the question of whether the algorithm works from the question of whether
steered activations are invertible. The L2 ≈ 0 recovery on natural activations makes the
large distances on steered activations meaningfully interpretable rather than just
negative results.

3. The paper is consistent across three model families. Llama, Qwen, and Gemma have
meaningfully different architectures and training pipelines, and the qualitative pattern
holds across all three. That consistency is real evidence, not cherry-picking.

4. The "Here attack" in Section F is the sleeper hit of the paper. A single-token
black-box suffix achieving comparable attack success rates to steering — while being
mechanistically completely different — is a vivid and concrete demonstration of the
threat-model separation the paper is arguing for. It also has direct practical
implications for how safety benchmarks should be designed.

5. The authors know what their experiments can and cannot establish, and they say so
clearly. Empirically proving non-existence over an exponentially large prompt space is
intractable by definition; the paper acknowledges this and frames the experiments
correctly as corroborating evidence rather than proof. That epistemic discipline is
rarer than it should be.

**Suggestions:**

SUGGESTIONS
------------
1. The theoretical argument has a significant dependency on Nikolaou et al. (2025),
which is currently an unreviewed arXiv preprint. The injectivity result that this paper
builds on is the load-bearing pillar of Theorems B.2 and B.3. The paper would be
stronger if it explicitly acknowledged this dependency and briefly discussed what
happens to the argument if injectivity is only approximate in practice — which may well
be the case for heavily fine-tuned instruct models where the parameter distribution is
far from Gaussian.

2. The "almost surely" result holds with probability one over random parameter
initializations, extended to trained models via "finite GD steps preserving injectivity."
But modern instruct models are trained through pretraining, SFT, RLHF, DPO, and other
alignment stages — calling that "a finite number of GD steps from a Gaussian init" is
technically true but practically misleading. A more honest treatment of where this
assumption could strain or break down would significantly strengthen the paper's
credibility with safety-focused readers.

3. All experiments use models at or below 1B parameters, and the acknowledgment of
this in the limitations section is too brief. Scaling behavior in LLMs is famously
unpredictable, and the audience most interested in this paper's conclusions — safety
researchers working on large deployed models — will want at least a theoretical argument
for why the result should or should not be scale-dependent. Even a brief discussion
would help.

4. The coefficient sweep results (Figures 2 and 7) show a striking pattern: even at
high λ, steered activations project back onto the original prompt under nearest-neighbor
reconstruction rather than drifting toward any other natural prompt. This seems like an
important geometric observation — steering moves activations into empty space, not
toward other reachable points — but the paper underanalyzes it. Unpacking this finding
more explicitly would strengthen the contribution.

5. The paper focuses entirely on discrete prompts. It would be worth at least briefly
addressing whether the non-surjectivity result extends to, or breaks down for, learned
continuous prompt embeddings and soft prompts, since these occupy a middle ground
between steering and black-box prompting that is relevant to some interpretability
methods.

 "All models are wrong, but some are useful."
                                                  — George E. P. Box

---

### Official Review · Reviewer_cnCC · 2026-02-27

**Fit:** 3
**Significance:** 2
**Confidence:** 2

**Summary:**

The paper studies whether activation steering replicated by prompting (ie. are steered activations realizable by some text prompt). Theoretically, the authors argue that prompt-induced activations occupy a sparse subset of the continuous activation space, implying that a generic steering perturbation will almost surely move the model to an activation state with no prompt preimage. They further prove an “almost sure sequence divergence” result, which states that even if a steered trajectory happens to intersect a naturally reachable activation at some step, the subsequent steps will almost surely diverge. Empirically, the authors test these claims on three small open-weight instruction-tuned language models under refusal and persona steering vectors, using an exhaustive token search method and ICL as prompt-based strategies for approximating the steering vector behavior. This work concludes that white-box controllability via activation interventions should not be conflated with black-box prompt vulnerability.

**Strengths:**

* The paper is well-written and easy to follow, and motivates why prompt-based and activation-based control should be treated as distinct threat methods well.
* This work presents both theoretical and empirical results, with experimental setup sweeping over steering settings and multiple open-weight instruction-tuned models.
* The two tested empirical methods complement each other; the ICL experiments provide particular evidence for not equating behavioral similarity with mechanistic equivalence, as even when ICL gets similar outcomes, the internal activations look different. In a similar vein, even exhaustive token search fails on steered activations.

**Suggestions:**

* I believe exact preimage is a very strong notion and the proofs mostly address equality in activation space, while many practical questions care about "equivalence classes" (eg. similar logit distributions). I wonder if a similar result can be quantified with eg. KL divergence between next-token distributions induced by steering vs by a prompt.
* Using SIPIT and many-context ICL limits the authors to smaller models, I'm wondering if they've tried methods like gradient-based prompt optimization.
* To have a more concrete empirical analysis of their theoretical intuition of random steering vectors moving off the natural manifold, I wonder if collecting natural activations for a given layer on a dataset of prompts/responses, fitting PCA, and checking the distances between a held-out natural activation set and a set of steered activations would be insightful?

---

### Meta-Review · Area_Chair_WiPm · 2026-03-01

**Recommendation:** Accept

**Metareview:**

This work is well motivated and provides compelling empirical and theoretical results. One weakness that Reviewer Zop1 mentioned that I'd like to reiterate is that the authors should make sure they do not over-claim with respect to their results. I suggest the authors read reviewers' feedback and take it into account in the camera ready version.

---

### Decision · Program_Chairs · 2026-03-02

Accept